# Mind the Gap: Discrepancy-Centric Diffusion for Aerial Image Reasoning and Synthesis

## Abstract

Diffusion models have emerged as powerful generators in text-to-image synthesis, yet their extension to aerial imagery remains limited due to unique challenges such as high object density and geometric distortions. In this paper, we propose MIND, a **M**ult**I**-scale discrepa**N**cy-centric latent **D**iffusion framework designed to address these issues and enable high-fidelity, semantically coherent aerial image synthesis. MIND introduces a theoretically justified method to estimate discrepancy maps that identify semantic and structural inconsistencies during generation, guiding both image synthesis and textual supervision. We incorporate these maps into the generation pipeline via three complementary mechanisms: (1) *actor-critic* visual reasoning to produce rationale-rich textual guidance using large language models, (2) discrepancy-augmented latent representation learning for spatial refinement, and (3) adaptive denoising that dynamically attends to "hard-to-learn" regions. Extensive experiments on VisDrone-DET and DroneRGBT demonstrate that MIND significantly outperforms state-of-the-art baselines in terms of visual quality, spatial alignment, and text-image consistency, establishing a strong foundation for structured and controllable aerial image synthesis.

## 1 Introduction

Aerial imagery is increasingly critical for applications such as urban planning, disaster response, and environmental monitoring (Zhang et al., 2022; Liu et al., 2022; Qu et al., 2024; Chang et al., 2023), creating a growing demand for large-scale, diverse, and high-quality data. Existing public aerial datasets (Cao et al., 2021; Xia et al., 2018), while valuable, suffer from limited geographic diversity, fixed viewpoints, and restricted environmental variability. These constraints hinder robust model training and simulation of dynamic real-world scenarios. As such, generating realistic and contextually consistent aerial images under diverse conditions has emerged as an essential alternative to fill coverage gaps, improve generalization, and support downstream vision tasks.

Diffusion models (Dhariwal & Nichol, 2021; Rombach et al., 2022) have recently advanced text-to-image generation by reversing a noising process to synthesize high-fidelity and semantically aligned outputs. Compared to generative adversarial networks (GANs) (Saxena & Cao, 2021), they offer better image fidelity, diversity, and controllability, and have been successfully applied to translation (Gao et al., 2024a), inpainting (Zhu et al., 2024), and restoration (Xia et al., 2023), as well as high-impact domains such as autonomous driving (Gao et al., 2024b; Chen et al., 2024b) and medical imaging (Zhan et al., 2024; Takagi & Nishimoto, 2023). Despite these advances, extending diffusion models to *text-to-aerial image synthesis* remains largely unexplored.

Direct adaptation of diffusion models to aerial imagery faces three key challenges: (1) aerial scenes feature many small objects whose details are easily lost during downsampling (Zang et al., 2024); (2) high object density and occlusion complicate spatial relationship modeling (Cao et al., 2021), making it difficult to preserve positional accuracy; and (3) large-scale, top-down perspectives introduce geometric distortions and scale variations that disrupt structural consistency. These factors often result in spatial misalignment, visual artifacts, and the omission of fine-grained structures in existing methods (Wang et al., 2023; Zhu et al., 2023; Ma et al., 2024). These challenges are further compounded by the lack of paired text–aerial image data, hampering the training of text-conditioned diffusion. While large language models (LLMs) (Wei et al., 2022b; Kojima et al., 2022) can produce

descriptions in a zero-shot manner, their outputs for aerial scenes are often ambiguous or imprecise, reducing their reliability as conditioning signals.

To address these challenges, we propose MIND, a **M**ult**I**-scale discrepa**N**cy-centric latent **D**iffusion framework for aerial image synthesis, aimed at enhancing its fidelity, consistency, and interpretability. The core idea is to innovatively exploit *discrepancy maps*, which capture semantic and structural inconsistencies between source images (or intermediate synthesis results) and their reconstructions, and use them as corrective signals to systematically localize failure regions and refine outputs in a targeted and informed manner. MIND proceeds with four complementary components: (1) discrepancy modeling for spatial error localization, (2) discrepancy-aware visual reasoning that enriches text conditioning with rationale-rich annotations highlighting generation gaps, (3) latent representation augmentation that injects spatially contextualized corrections, and (4) adaptive denoising that dynamically attends to uncertain regions during later diffusion steps.

Specifically, discrepancy maps are estimated through a hierarchical, attention-modulated mechanism grounded in reconstruction residuals, which we theoretically justify as a principled way to capture global and local structured inconsistencies using multi-scale design. Within an *actor-critic* scheme, the maps help LLMs produce descriptions and rationales that explicitly interpret regions where the model struggles, thereby improving semantic grounding. During diffusion, the maps serve dual roles: augmenting latent representations to recover missing details and dynamically adjusting conditioning vectors to refine unresolved inconsistencies. By conditioning generation on integrated image-text-discrepancy embeddings, MIND effectively enhances interpretability, preserves spatial coherence, and achieves high-fidelity synthesis in complex aerial environments. Experiments on VisDrone-DET (Cao et al., 2021) and DroneRGB-T (Peng et al., 2020) demonstrate consistent improvements over state-of-the-art baselines in fidelity, alignment, and text–image consistency. Our major contributions are summarized as follows:

- A theoretically justified method to model discrepancy maps from reconstruction residuals, capturing inconsistencies across multiple spatial scales.
- An actor-critic visual reasoning paradigm that uses discrepancy cues to elicit rationale-rich textual guidance and address the lack of paired data for experimental evaluation and future research.
- An adaptive denoising strategy that dynamically corrects error-prone regions that improves spatial fidelity and scene alignment.
- A unified discrepancy-centric diffusion framework MIND that integrates technic components for interpretable, high-fidelity aerial image synthesis.

## 2 RELATED WORK

**Text-guided image generation**. Text-guided image generation has progressed rapidly, evolving from early retrieval and template-based methods to modern deep generative models capable of producing semantically aligned, high-fidelity images. Early text-to-image methods relied on optimization-based techniques, where pretrained vision-language models (e.g., CLIP (Radford et al., 2021) and BLIP (Li et al., 2022)) guided pixel-level updates via handcrafted or learned losses (Liang et al., 2024; Mahajan et al., 2024). These approaches often yielded low-detail or artifact-prone images due to the lack of a dedicated generative backbone. The advent of autoregressive (Pan et al., 2024; Qi et al., 2023) and diffusion-based models (Ho et al., 2020; Rombach et al., 2022; Ma et al., 2024; Karras et al., 2024; Meral et al., 2024) marked a major shift. Diffusion models enable image generation with enhanced diversity, semantic alignment, and controllability. Latent diffusion (Rombach et al., 2022) further improves efficiency by operating in a compressed latent space, allowing high-resolution synthesis at reduced computational cost. While diffusion models have demonstrated strong capabilities in generative vision tasks, existing research on aerial imagery has primarily focused on semantic segmentation (Liu et al., 2024), object detection (Li et al., 2023; Chen et al., 2024a), and domain-specific translation tasks (e.g., aerial-to-map or map-to-aerial conversion)(Fu et al., 2021). Although models such as CycleGAN (Zhu et al., 2017), Pix2Pix (Isola et al., 2017), and Pix2PixHD (Wang et al., 2018) are effective in preserving structural integrity and semantic content, they remain limited by their reliance on *fixed input–output mappings* and lack the flexibility needed for open-ended, *text-guided aerial image synthesis*.

**Knowledge integration for visual reasoning**. Enhancing the multi-modal reasoning capabilities of LLMs has become an active area of research, with prompt-based techniques demonstrating ef-

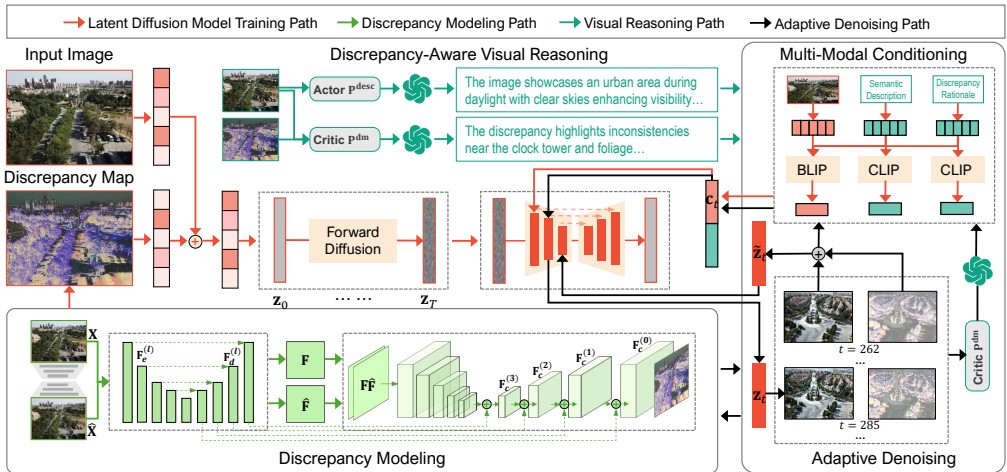

Figure 1: An overview of **MIND** that first estimates discrepancy map, which is used for visual reasoning, augmenting representations, and modulating multi-modal conditioning for adaptive denoising to generate complex aerial images.

fectiveness in inducing structured reasoning across modalities. Existing approaches can be broadly categorized into three types: (1) zero-shot and few-shot prompting, which introduce intermediate reasoning steps (Wei et al., 2022b; Perez et al., 2021); (2) Chain-of-Thought (CoT) prompting, which decomposes reasoning into step-wise semantic steps (Wei et al., 2022a; Zhang et al., 2024; Zellers et al., 2019; Marino et al., 2019; Mao et al., 2023), demonstrating strong performance in improving reasoning accuracy; and (3) Tree-of-Thought (ToT) prompting, which generalizes CoT by enabling multi-path exploration (Yao et al., 2023). However, due to the unique characteristics of aerial imagery such as dense object distributions, geometric distortions, and structural complexity, language models often produce verbose and imprecise descriptions that fail to accurately capture the underlying semantics when not guided by a carefully designed prompting.

## 3 DESIGN OF MIND

In this section, we present **MIND** with technical details, the architecture of which is illustrated in Figure 1, providing a high-level overview of how discrepancy maps are estimated, interpreted, and leveraged throughout the generation process.

### 3.1 DISCREPANCY MODELING

We define discrepancy as localized deviations between the reconstructed outputs and their expected visual features, often manifesting as spatial misalignments, missing details, or unnatural artifacts. Detecting and correcting such inconsistencies is essential for highlighting hard-to-learn regions and producing high-fidelity images, especially in dense and fine-grained aerial scenes. To justify our design, we first show that reconstruction residuals provide a principled signal for discrepancy:

**Theorem 1** *Let $\mathcal{M}$ be the manifold of valid images, and $\widehat{\mathbf{X}} = \mathcal{D}(\mathcal{E}(\mathbf{X}))$ the autoencoder reconstruction of $\mathbf{X}$. If the residual $\mathbf{R}(\mathbf{X}) = \mathbf{X} - \widehat{\mathbf{X}}$, then under standard assumptions of manifold projection and local linearity:*

$$\|\mathbf{R}(\mathbf{X})\| = \mathrm{dist}(\mathbf{X}, \mathcal{M}) + o(\|\mathbf{X} - \mathbf{X}_0\|), \qquad \mathbf{R}(\mathbf{X}) = \Pi_{T_{\mathbf{X}_0}\mathcal{M}^\perp}(\mathbf{X} - \mathbf{X}_0) + o(\|\mathbf{X} - \mathbf{X}_0\|), \quad (1)$$

*where $T_{\mathbf{X}_0}\mathcal{M}$ is the tangent space of $\mathcal{M}$ at $\mathbf{X}_0$ and $\Pi_{T_{\mathbf{X}_0}\mathcal{M}^\perp}$ denotes orthogonal projection onto its normal space. Thus, residuals $\mathbf{R}(\mathbf{X})$ suppress on-manifold (tangent-space) variations and emphasize off-manifold (normal-space) structural deviations.*

Theorem 1 implies that off-manifold inconsistencies (e.g., misalignments, missing objects, artifacts) persist in $\mathbf{R}(\mathbf{X})$, while on-manifold variations (e.g., style, illumination) vanish to first order. This justifies that $\mathbf{R}(\mathbf{X})$ serves as a principled discrepancy signal. Building on this, we argue that hierarchical modeling provides a natural way to capture discrepancies:

**Proposition 1** *Let $\mathbf{R}(\mathbf{X})$ be decomposed into multi-scale components $\{\mathbf{R}_s\}_{s=0}^{S}$ by a Parseval frame (e.g., wavelets or Laplacian pyramid). Then*

$$\|\mathbf{R}(\mathbf{X})\|_2^2 = \sum_{s=0}^{S} \|\mathbf{R}_s\|_2^2, \tag{2}$$

*with coarse scales concentrating global layout errors and fine scales capturing local object omissions. Moreover, subband energies are stable under small geometric deformations, ensuring robust detection of discrepancies across scales.*

Proposition 1 indicates that discrepancies manifest across both coarse and fine scales, while remaining stable under small deformations. Proofs of Theorem 1 and Proposition 1 are provided in the Appendix A.4 and A.5. Motivated by these analyses, we design a multi-scale discrepancy module, we develop a multi-scale discrepancy estimation module (Figure 9) that jointly reasons over local details and global layouts, ensuring spatial integrity and semantic consistency in aerial synthesis.

**Multi-scale spatial feature extraction**. Given an input image $\mathbf{X}$, we first obtain its reconstruction $\hat{\mathbf{X}}$ from a self-trained autoencoder, which serves as a contrastive reference for discrepancy modeling. Both $\mathbf{X}$ and $\hat{\mathbf{X}}$ are processed by two parallel encoder–decoder networks with identical architecture, extracting multi-scale spatial features while preserving structural integrity and contextual consistency. Each encoder $\mathcal{E}_{\text{ms}}(\cdot)$ consists of five convolutional layers with downsampling by a factor of 2 per layer, producing feature maps at scales $\frac{1}{2}$ to $\frac{1}{32}$ of the original resolution:

$$\mathcal{E}_{\text{ms}}(\mathbf{X}) = \{\mathbf{F}_e^{(l)}\}_{l=1}^{5}, \quad \mathbf{F}_e^{(l)} \in \mathbb{R}^{C_l \times \frac{H}{2^l} \times \frac{W}{2^l}}. \tag{3}$$

The decoder mirrors the encoder, progressively upsampling features with skip connections:

$$\mathbf{F}_d^{(l)} = \text{Upsample}(\mathbf{F}_d^{(l+1)}) + \mathbf{F}_e^{(l)}, \ \forall l \in \{1, \ldots, 4\}, \ \mathbf{F}_d^{(5)} = \mathbf{F}_e^{(5)}. \tag{4}$$

This yields two sets of features, $\mathbf{F}$ and $\hat{\mathbf{F}}$, representing $\mathbf{X}$ and $\hat{\mathbf{X}}$, respectively.

**Discrepancy map estimation**. To estimate discrepancies, we concatenate $\mathbf{F}$ and $\hat{\mathbf{F}}$ along the channel dimension and pass the result through a dedicated encoder–decoder network. During decoding, multi-scale features $\mathbf{F}_d^{(l)}$ ($l \in [1, 4]$) are injected into the corresponding layers to refine discrepancy-sensitive features with contextual information. Instead of simple addition or concatenation, we adopt an attention-based fusion (Woo et al., 2018), enabling adaptive integration of spatial and discrepancy cues. Let $\mathbf{F}_c^{(l)}$ be the discrepancy features at decoding layer $l$. Attention weights are computed as:

$$\mathbf{W}^{(l)} = \sigma\left(\text{Conv}\left(\mathbf{F}_c^{(l)} + \text{Upsample}(\mathbf{F}_d^{(l)})\right)\right), \tag{5}$$

and the fused features are

$$\mathbf{F}_c^{(l-1)} = \text{Upsample}\left(\mathbf{W}^{(l)} \odot \text{Upsample}(\mathbf{F}_d^{(l)}) + \mathbf{F}_c^{(l)}\right), \tag{6}$$

where $\odot$ denotes element-wise product. Finally, a convolution applied to $\mathbf{F}_c^{(0)}$ predicts the discrepancy map $\mathbf{D} \in \mathbb{R}^{3 \times H \times W}$.

**Optimization**. We minimize a joint loss that encourages accurate localization and quantification of spatial differences between $\mathbf{X}$ and $\hat{\mathbf{X}}$. Let $\mathbf{D}_{\text{gt}} = |\mathbf{X} - \hat{\mathbf{X}}|$ be the ground-truth residual map and $\delta = \mathbf{D} - \mathbf{D}_{\text{gt}}$ the estimation error. We define *discrepancy loss* and *reconstruction loss* as follows:

$$\mathcal{L}_\delta = \|\delta\|_1, \qquad \mathcal{L}_{\text{rec}} = \|\mathbf{X} - \hat{\mathbf{X}}\|_2^2, \tag{7}$$

The total loss is computed as their weighted combination:

$$\mathcal{L}_{\text{dm}} = \lambda_\delta \, \mathcal{L}_\delta + \lambda_{\text{rec}} \, \mathcal{L}_{\text{rec}}, \tag{8}$$

with $\lambda_\delta, \lambda_{\text{rec}}$ controlling the balance. This formulation enables spatially localized refinement of discrepancy signals, preserves both global layout and fine details, and highlights error-prone regions critical for aerial image synthesis.

## 3.2 ACTOR-CRITIC VISUAL REASONING

A core obstacle in text-to-aerial image synthesis is the lack of paired text-image datasets, which prevents diffusion models from learning reliable semantic guidance. while LLMs with Chain-of-Thought (CoT) prompting (Wei et al., 2022b; Kojima et al., 2022) can generate scene descriptions in a zero-shot manner, the inherent complexity of aerial imagery often leads LLMs to produce ambiguous or imprecise outputs. To mitigate this, we propose an *actor-critic prompting paradigm* that integrates image description with discrepancy-aware refinement: the *actor* generates initial semantic descriptions, and the *critic* refines them through localized visual reasoning to highlight uncertain regions, correct omissions, and enhance textual grounding.

- *Actor (Scene-level describing)*. Givne an aerial image $\mathbf{X} \in \mathcal{X}$ and its object list $\mathcal{O}$, we define a CoT prompt $P^{\text{desc}}$ to guide the LLM to generate a semantic description:

$$G = \text{LLM}(P^{\text{desc}}; \mathbf{X}, \mathcal{O}) \qquad (9)$$

$P^{\text{desc}}$ is curated to elicit high-level scene overview, such as viewpoint, environmental conditions, and spatial composition, serving as a global semantic prior for diffusion-based synthesis.

> $P^{\text{desc}}$: Analyze an *[aerial image]* with objects *[$o_1, o_2, ..., o_k$]*, and generate a precise yet rich description that captures key visual elements, including both prominent and subtle visual details, time of day and lighting conditions, and spatial layout. Consider their interactions within the scene to form a cohesive, structured depiction that enhances spatial and contextual understanding.

- *Critic (Localized reasoning)*. To refine under-specified or ambiguous regions in $G$, we form a discrepancy-aware prompt $P^{\text{dm}}$ by combining the discrepancy map with the image $\mathbf{X}$ and coarse description $G$ to produce a rationale-rich refinement:

$$\hat{G} = \text{LLM}(P^{\text{dm}}; \mathbf{X}, \mathbf{D}, G) \qquad (10)$$

This emphasizes regions with high uncertainty, particularly those that are difficult to learn or prone to generation errors, guiding the model toward more precise semantic grounding.

> $P^{\text{dm}}$: Generate a concise rationale that examines an *[aerial image]* and its *[scene-level description]* and *[discrepancy map]* to uncover reconstruction inconsistencies, and analyze the spatial composition, focusing on how discrepancies affect spatial coherence, interact across different areas, influencing depth perception, structural continuity, and overall scene interpretation.

### 3.3 LATENT DIFFUSION USING ADAPTIVE DENOISING

Diffusion models (Appendix A.1) operate in latent space to improve efficiency. An input image $\mathbf{X} \in \mathcal{X}$ is encoded into a latent representation $\mathbf{z}_0$ via a VAE, then progressively corrupted into $\mathbf{z}_t$ over timesteps $t \in [1, T]$ by a fixed noise schedule. A denoising model $\epsilon\theta(\mathbf{z}_t, t, \mathbf{c}_t)$ is trained to predict the injected noise $\epsilon$ given conditioning $\mathbf{c}_t$. In MIND, we enhance this process by injecting discrepancy-driven corrections to improve spatial fidelity and semantic alignment.

**Latent representation augmentation**. Compression in latent space often suppresses fine-grained details such as small objects or local alignments. To restore these, we design a *spatial attention-modulated fusion* to inject corrective signals from the discrepancy map into latent representation. Let $\mathbf{z}_d$ be the latent features of $\mathbf{X}$'s discrepancy map, a spatial attention weight $\boldsymbol{\alpha}$ is computed as $\boldsymbol{\alpha} = \sigma(\text{Conv}(\mathbf{z}_d))$. The augmented latent representation $\widetilde{\mathbf{z}}$ is then derived as:

$$\widetilde{\mathbf{z}} = \mathbf{z} + \mathbf{z}_d \odot \boldsymbol{\alpha} \qquad (11)$$

which ensures that discrepancy-sensitive regions are emphasized while preserving global coherence.

**Conditioning vector construction**. We initialize diffusion with $\mathbf{z}_0 = \widetilde{\mathbf{z}}$ and construct the conditioning vector $\mathbf{c}$ that fuses three complementary modalities: (1) the original image $\mathbf{X}$ for spatial priors and layout; (2) the global semantic description $G$ from the actor; and (3) the rationale $\hat{G}$ from the critic. We employ BLIP (Li et al., 2022) to jointly embed $(\mathbf{X}, G)$ and $(\mathbf{X}, \hat{G})$, and further encode $G$ and $\hat{G}$ independently using CLIP (Radford et al., 2021) to preserve distinct and nuanced semantics that may be diluted in a unified representation. The resulting embeddings are concatenated to form the conditioning vector $\mathbf{c}$, which integrates both global understanding and localized corrective context for the initial denoising process.

**Adaptive denoising**. Standard diffusion $\epsilon_\theta(\mathbf{z}_t, t, \mathbf{c}_t)$ follows a deterministic trajectory $\{\mathbf{z}_t\}_{t=1}^{T}$, where $\mathbf{z}_t$ is obtained by reversing the forward noise process, limiting adaptation to emerging inconsistencies. To improve generative fidelity, our objective is to dynamically steer this trajectory using discrepancy-guided signals. To this end, at each denoising timestep $t$, a partially denoised image $\mathbf{X}_t$ decoded from $\mathbf{z}_t$ is passed through our discrepancy estimation network to compute its discrepancy map $\mathbf{D}_t$, which serves as an adaptive calibration signal for denoising in the following ways:

- *Latent correction*. $\mathbf{D}_t$ is projected into latent space and injected into the current latent $\mathbf{z}_t$ using the spatial attention-modulated fusion, resulting in an updated latent vector $\widetilde{\mathbf{z}}_t$ that incorporates spatial corrections before the next denoising step.

- *Conditioning update.* The refined latent $\widetilde{\mathbf{z}}_t$ is used in place of the original when recomputing cross-attention between visual-textual modalities:

$$\mathbf{c}_t = g(\widetilde{\mathbf{z}}_t, G, \hat{G}) \tag{12}$$

where $g(\cdot)$ denotes the aforementioned multimodal fusion operator combining augmented latent $\widetilde{\mathbf{z}}_t$ with static texts $G$ and $\hat{G}$. This dynamic conditioning ensures temporal consistency and enables the model to incorporate recent visual feedback during generation.

By continuously refining both the latent features and semantic conditioning with discrepancy feedback, the model adaptively prioritizes uncertain or hard-to-learn regions, enabling the denoising process to better restore spatial details throughout generation. To justify adaptive denoising, we note that the reliability of discrepancy maps depends on the noise variance $\sigma_t^2$ at timestep $t$.

**Proposition 2** *Let $\mathbf{D}_t$ be the discrepancy estimated from a partially denoised image $\mathbf{X}_t$. Assume $\mathbf{D}_t$ has bias $\mathcal{O}(\sigma_t)$ and variance $\mathcal{O}(\sigma_t^2)$ with respect to the true residual. Then discrepancy-guided updates of the latent $\mathbf{z}_t$ decrease denoising error for all $t$ such that $\sigma_t \leq \sigma^\star$, for some threshold $\sigma^\star$. For $t$ with $\sigma_t \gg \sigma^\star$, the updates may be dominated by estimation noise.*

Proof of Proposition 2 is provided in the Appendix A.6. This implies that discrepancy maps provide effective corrective signals primarily in later timesteps, where the noise level is low and reconstructions are reliable. Accordingly, MIND applies adaptive denoising only after $t \gtrsim T/2$.

## 3.4 Training and Inference

MIND is trained end-to-end using the standard diffusion objective, extended to incorporate discrepancy-driven updates of both latents and conditioning. The training loss is defined as:

$$\mathcal{L}_{\text{MIND}} = \mathbb{E}_{\mathbf{z}_0, \epsilon \sim \mathcal{N}(0,\mathbf{I}), t} \left[ \|\epsilon - \epsilon_\theta(\mathbf{z}_t, t, \mathbf{c}_t)\|_2^2 \right] \tag{13}$$

where $\mathbf{z}_t$ and $\mathbf{c}_t$ are discrepancy-aware noisy latent and conditioning vector at timestep $t$. Both the denoising parameters $\theta$ and the auxiliary modules for updating $\mathbf{z}_t$ and $\mathbf{c}_t$ are jointly optimized. At inference time, MIND starts from Gaussian noise and iteratively predicts a clean latent $\hat{\mathbf{z}}_0$ under adaptive denoising. The final output image is reconstructed via the VAE decoder $\mathcal{D}(\hat{\mathbf{z}}_0)$.

# 4 Experimental Evaluation and Analysis

## 4.1 Experimental Setup

**Datasets.** We use two publicly available aerial datasets: (1) *VisDrone-DET* (Cao et al., 2021), featuring high-density daytime urban scenes, and (2) *DroneRGB-T* (Peng et al., 2020), which includes challenging low-light and multimodal conditions. They provide complementary coverage for assessing fidelity and semantic robustness.

**Implementation.** Images are resized to $512 \times 512 \times 3$ and encoded into a latent space via a custom-trained autoencoder. MIND is trained for 60 epochs on both datasets. Gaussian noise is injected using the DDPM scheduler (Ho et al., 2020) with $\beta$ linearly increasing from 0.001 to 0.012 over 1000 steps. Generation employs DDIM (Song et al., 2020) with 300 denoising steps and a guidance scale of 8.0. GPT-4o is used for actor–critic reasoning. Discrepancy modeling is supervised with a joint loss, where $\lambda_{\text{rec}} = 1.2$ and $\lambda_\delta = 1.0$ balance reconstruction and discrepancy objectives.

**Metrics.** Visual quality is assessed with four standard metrics: (1) Fréchet Inception Distance (FID) (Heusel et al., 2017) for distributional similarity; (2) Peak Signal-to-Noise Ratio (PSNR) (Wang et al., 2004) for pixel-level fidelity; (3) Kernel Inception Distance (KID) (Binkowski et al., 2018) for unbiased feature distribution deviation, especially suitable for smaller datasets; and (4) Learned Perceptual Image Patch Similarity (LPIPS) (Zhang et al., 2018) for perceptual alignment with human vision. To evaluate actor-critic visual reasoning, we adopt a two-phase protocol: (1) CLIP Score to measure text–image alignment between generated descriptions and aerial inputs, and (2) FID/LPIPS to compare image quality under different prompting strategies.

Table 1: Visual performance across *VisDrone-DET* (Cao et al., 2021) and *DroneRGB-T* (Peng et al., 2020). Results are based on 3,300 generated samples for VisDrone-DET and 1,807 for DroneRGB-T. **Bold** indicates the best results, while underline represents the second-best.

| Models | FID ↓ | | PSNR ↑ | | KID ↓ | | LPIPS ↓ | |
| --- | --- | --- | --- | --- | --- | --- | --- | --- |
| | VisDrone-DET | DroneRGB-T | VisDrone-DET | DroneRGB-T | VisDrone-DET | DroneRGB-T | VisDrone-DET | DroneRGB-T |
| ARLDM (Pan et al., 2024) | 106.30 | 199.51 | 6.60 | 6.77 | 0.06 | 0.21 | 0.60 | 0.64 |
| Conform (Meral et al., 2024) | 110.96 | 184.17 | 6.15 | 7.04 | 0.06 | 0.17 | 0.58 | 0.62 |
| Make-a-Scene (Gafni et al., 2022) | 139.81 | 212.29 | 5.70 | 5.88 | 0.07 | 0.24 | 0.64 | 0.68 |
| Stable Diffusion (SD) (Rombach et al., 2022) | 101.05 | 182.23 | 5.93 | 6.97 | 0.06 | 0.17 | 0.57 | 0.61 |
| Versatile Diffusion (VD) (Xu et al., 2023) | 102.18 | 188.16 | 5.77 | 6.26 | 0.07 | 0.19 | 0.58 | 0.62 |
| DiffusionSAT (Khanna et al., 2024) | 80.98 | 183.88 | **7.08** | **7.19** | **0.05** | **0.14** | 0.61 | 0.63 |
| AeroGen (Tang et al., 2025) | 86.94 | 184.03 | 7.01 | 7.06 | **0.05** | 0.16 | 0.62 | 0.63 |
| Average | 104.03 | 190.61 | 6.32 | 6.74 | 0.06 | 0.18 | 0.60 | 0.63 |
| MIND (w/o AD) | 83.76 | 172.22 | 6.92 | 7.07 | **0.05** | 0.16 | 0.51 | 0.57 |
| MIND (ours) | **78.02** (-24.9%) | **158.18** (-17.0%) | 7.03 (+11.2%) | 7.14 (+5.9%) | **0.05** (-16.7%) | 0.16 (-12.6%) | **0.47** (-21.7%) | **0.53** (-16.3%) |

| Source Images | Ours | SD | ARLDM | Conform | DiffusionSat |
| --- | --- | --- | --- | --- | --- |

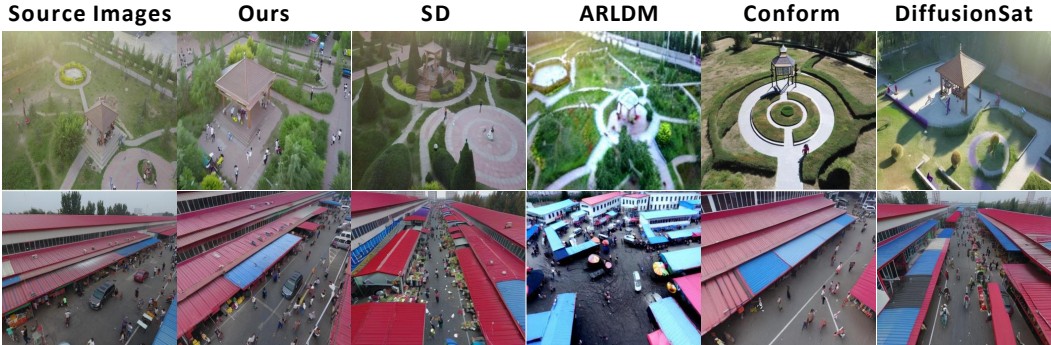

Figure 2: Visual performance comparison on VisDrone-DET.

| Source Images | Ours | SD | Make a scene | VD | DiffusionSat |
| --- | --- | --- | --- | --- | --- |

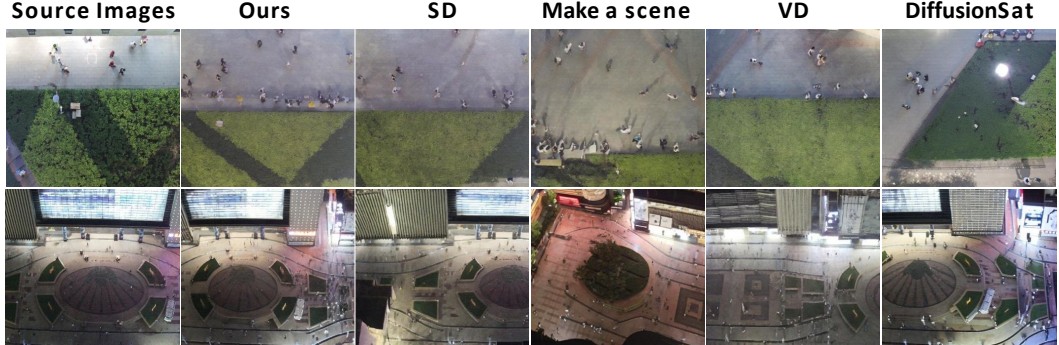

Figure 3: Visual performance comparison on DroneRGB-T.

## 4.2 EVALUATION ON AERIAL IMAGE SYNTHESIS

**Performance overview**. Table 1 compares MIND with seven state-of-the-art baselines (five general-purpose and two aerial-specific diffusion models) on *VisDrone-DET* (Cao et al., 2021) and *DroneRGB-T* (Peng et al., 2020). MIND achieves consistent gains across FID, PSNR, KID, and LPIPS, confirming its ability to generate high-fidelity, semantically aligned aerial imagery. The ablation variant MIND (w/o AD) isolates the contribution of the adaptive denoising module.

**Visualization comparison**. Figures 2 and 3 present randomly sampled synthesized aerial images from both datasets. Baselines often fail to preserve spatial layouts and coherence to the source images, showing blur or distort structures (e.g., radial gardens or roof patterns). In contrast, MIND reconstructs them with higher fidelity. Under varied illumination, it robustly maintains scene geometry and accurately models shadow boundaries and object contours, enhancing realism.

**Qualitative analysis of visual fidelity**. We further showcase diverse synthesized examples generated by MIND and conduct an in-depth qualitative analysis across three dimensions.

- *Geometric structure, spatial layout, and object detail preservation*. Figure 4 compares sparse-object natural scenes and dense-object urban scenes. In cluttered natural settings, baselines often produce over-smoothed or distorted textures, whereas MIND preserves fine structures such as seasonal leaf variations. In dense urban layouts, it maintains small-object fidelity (e.g., pedes-

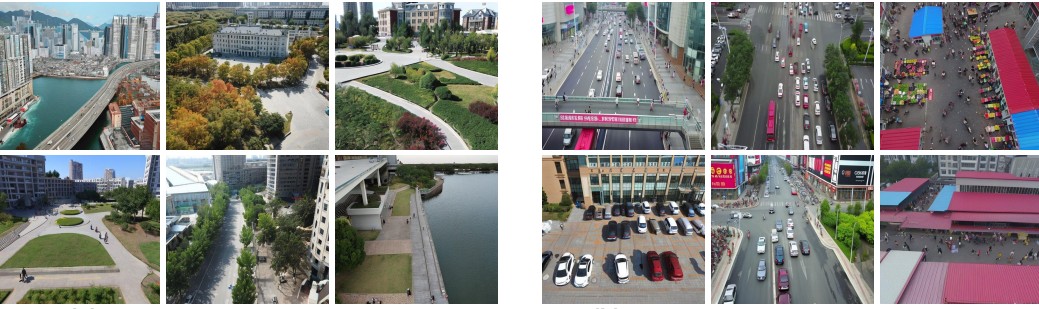

**(a)** Complex Backgrounds & Sparse Objects   **(b)** High-Density Small Object Distributions

Figure 4: Qualitative analysis of geometric structure, spatial layout, and object detail preservation.

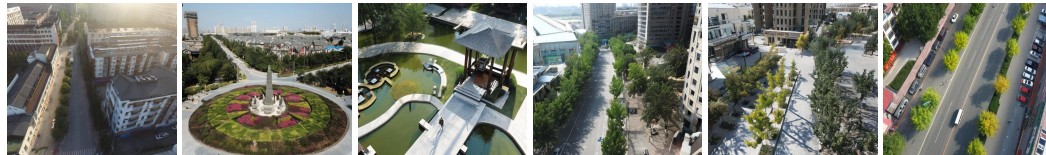

Figure 5: Capturing and modeling fine-grained illumination and shadow patterns.

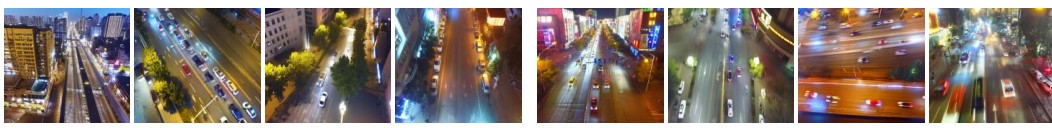

**(a)** Nighttime Environments   **(b)** Motion Blur Patterns

Figure 6: Image synthesis under high-noise conditions.

trians, vehicles) and directional consistency (e.g., cars oriented correctly on intersecting roads), demonstrating robustness to occlusion and geometric distortion.

- *Capturing and modeling fine-grained illumination and shadow patterns.* As shown in Figure 5, MIND captures localized lighting dynamics and produces directionally consistent shadows from buildings and trees. This enhances photorealism and provides geometric cues essential for aerial scene interpretation.

- *Robust image synthesis under high-noise conditions: nighttime environments and motion blur.* Low-light and high-motion settings pose major challenges for aerial synthesis. In Figure 6, MIND maintains structural fidelity while faithfully reproducing noise patterns and blur progression. This robustness is particularly valuable for downstream tasks such as surveillance and navigation, where realistic noise characteristics aid reliable scene understanding.

### 4.3 EVALUATION OF DISCREPANCY MAPS AND ACTOR-CRITIC REASONING

**Discrepancy maps**. We showcase a few random selected examples of generated discrepancy maps in Figure 7(a), including challenging nighttime scenes. These visualizations demonstrate that the maps not only localize structural misalignments and missing details but also remain effective under challenging conditions, providing reliable guidance for targeted refinement during synthesis.

**Actor-Critic reasoning**. We evaluate the effectiveness of the proposed actor-critic visual reasoning in Table 2, where *Image Describing* (serving as the actor) employs a CoT prompt ($P^{\text{desc}}$ introduced in Section 3.2) to generate semantic-level

Table 2: Performance on different reasoning strategies.

| Strategy | VisDrone-DET | | | DroneRGB-T | | |
|---|---|---|---|---|---|---|
| | CLIP ↑ | FID ↓ | LPIPS ↓ | CLIP ↑ | FID ↓ | LPIPS ↓ |
| Image Describing | 34.06 | 89.13 | 0.58 | 23.17 | 177.40 | 0.60 |
| Visual Reasoning | 35.97 | 84.27 | 0.54 | 25.29 | 174.86 | 0.59 |
| Actor-Critic | **36.09** | **78.02** | **0.47** | **26.24** | **158.18** | **0.53** |

description, *Visual Reasoning* ($P^{\text{visual}}$ introduced in Appendix A.7) relies solely on the input image and focuses on salient regions without capturing localized inconsistencies, and *Actor-Critic Reasoning* (two-stage prompting $P^{\text{desc}}$ and $P^{\text{dm}}$ introduced in Section 3.2) leverages the predicted discrepancy map to guide targeted refinement. As shown in Table 2, our actor–critic formulation achieves the best overall scores on both datasets, confirming that discrepancy-guided refinement provides complementary benefits beyond scene-level descriptions or purely visual reasoning. To further illustrate, we provide a case study to showcase the text generations from different prompts in Figure 7(b). These examples highlight that while $P^{\text{desc}}$ and $P^{\text{visual}}$ capture global semantics and

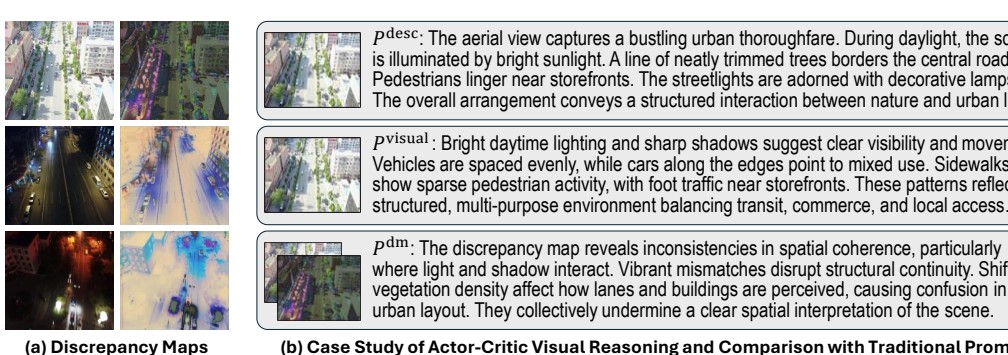

**(a) Discrepancy Maps**   **(b) Case Study of Actor-Critic Visual Reasoning and Comparison with Traditional Prompt**

Figure 7: Case study of generated discrepancy maps and actor-critic visual reasoning.

Table 3: Ablation study on *VisDrone-DET*.

| D-Reasoning | D-Denoising | FA | FID ↓ | PSNR ↑ | LPIPS ↓ |
|:---:|:---:|:---:|:---:|:---:|:---:|
| ✗ | ✗ | ✗ | 101.05 | 5.93 | 0.57 |
| ✓ | ✗ | ✗ | 94.40 | 6.95 | 0.57 |
| ✗ | ✓ | ✗ | 85.42 | 6.90 | 0.57 |
| ✗ | ✗ | ✓ | 88.28 | 7.51 | 0.58 |
| ✓ | ✗ | ✓ | 83.76 | 6.92 | 0.51 |
| ✓ | ✓ | ✓ | **78.02** | **7.03** | **0.47** |

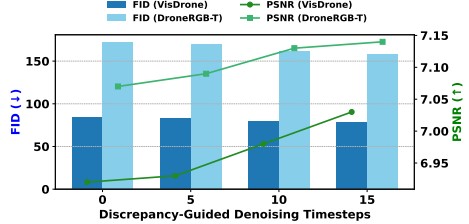

Figure 8: Evaluation on adaptive denoising.

coarse structures, only the discrepancy-aware prompt $P^{\mathrm{dm}}$ explicitly exposes subtle errors and inconsistencies. This demonstrates the advantage of our actor-critic reasoning, where the combination of $P^{\mathrm{desc}}$ and $P^{\mathrm{dm}}$ produces scene description and rationale that are both semantically rich and error-aware, yielding higher-quality guidance for aerial image synthesis.

## 4.4 ABLATION STUDY

We conduct an ablation study to evaluate the contribution of each module in MIND, focusing on discrepancy-aware reasoning (D-Reasoning), discrepancy-guided denoising (D-Denoising), and feature augmentation (FA), as summarized in Table 3. We also analyze the effect of applying discrepancy guidance at different stages of the denoising process, shown in Figure 8. Removing all components leads to the weakest results (FID 101.05, PSNR 5.93, LPIPS 0.57), consistent with the Stable Diffusion baseline. Introducing D-Reasoning alone gives modest gains, showing its standalone utility. Larger improvements arise from D-Denoising and FA, which significantly reduce FID and improve PSNR. The full model, combining all three modules, achieves the strongest performance (FID 78.02, PSNR 7.03, LPIPS 0.47), highlighting their complementary effects. Figure 8 further demonstrates that progressively injecting discrepancy cues during denoising leads to consistent improvements. As timesteps with discrepancy guidance increases (from 0 to 15), both FID decreases and PSNR increases, indicating enhanced image fidelity and reconstruction accuracy.

## 4.5 DISCUSSION AND LIMITATIONS

While MIND achieves notable improvements in semantic fidelity and spatial alignment, several challenges remain. The model struggles with complex texts involving multi-object interactions, where reasoning about fine-grained relationships is required. Performance also degrades in extremely dark scenes, where low illumination and high noise impair structural preservation and contextual coherence. Addressing these challenges, for example by incorporating stronger temporal priors or physics-aware constraints, is an important direction for future work.

## 5 CONCLUSION

This paper presents MIND, a novel discrepancy-centric latent diffusion framework for high-fidelity aerial image synthesis. By introducing multi-scale discrepancy maps and integrating them into the generation process through visual reasoning, latent representation augmentation, and adaptive denoising, MIND addresses the unique challenges of aerial imagery such as geometric distortion, dense object distributions, and semantic misalignment. Extensive experiments on VisDrone-DET and DroneRGBT demonstrate that MIND significantly outperforms state-of-the-art methods in visual quality, spatial consistency, and text-image alignment. This work establishes a robust foundation for structured, interpretable, and controllable generation in aerial visual understanding.

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

# A  APPENDIX

This appendix contains additional technical details and discussions of our submission as follows:

- A.1: A detailed overview of diffusion models, including forward, reverse, and inference processes in latent space.

- A.2: Formal problem description for aerial image synthesis with discrepancy-aware guidance.

- A.3: The detailed structure of the discrepancy modeling module.

- A.4: Proof of Theorem 1, establishing residuals as normal-space projections.

- A.5: Proof of Proposition 1, showing multi-scale decomposition and stability.

- A.6: Justification of Proposition 2, motivating adaptive denoising.

- A.7: Additional details on traditional visual reasoning prompts.

- A.8: Disclosure on the use of LLMs in this paper.

## A.1  DIFFUSION MODELS

The overall generation pipeline using diffusion models involves three core phases: a *forward process* that progressively corrupts clean data by adding noise, a *reverse process* that learns to denoise this noisy input, and an *inference process* that iteratively removes noise from a pure Gaussian sample to synthesize a structured output. To improve scalability and efficiency, modern diffusion models, such as Stable Diffusion Rombach et al. (2022), often operate in a latent space rather than directly in pixel space. More specifically, let $\mathbf{X}_0 \in \mathbb{R}^{3 \times H \times W}$ denote a clean image. This input image is first encoded into a compact latent representation $\mathbf{z}_0 = \mathcal{E}(\mathbf{X}_0)$ using a variational autoencoder (VAE) encoder $\mathcal{E}(\cdot)$ Cemgil et al. (2020).

**Forward process**. During the forward process, a predefined noise schedule is applied to progressively corrupt $\mathbf{z}_0$ over timesteps $t \in [1, T]$, producing noisy latent vectors $\mathbf{z}_t$:

$$\mathbf{z}_t = \sqrt{\bar{\alpha}_t}\mathbf{z}_0 + \sqrt{1 - \bar{\alpha}_t}\boldsymbol{\epsilon}, \quad \boldsymbol{\epsilon} \sim \mathcal{N}(0, \mathbf{I}) \tag{14}$$

where $\bar{\alpha}_t$ is the cumulative product of noise scaling coefficients that controls the signal-to-noise ratio at timestep $t$.

**Reverse process**. A denoising model $\boldsymbol{\epsilon}_\theta(\mathbf{z}_t, t, \mathbf{c})$, parameterized by $\theta$, is trained to predict and remove the added noise $\boldsymbol{\epsilon}$. The model is conditioned on external information $\mathbf{c}$, such as textual description or semantic layout, which provides high-level guidance for generating the desired output. The training objective minimizes the mean squared error between predicted and true noise:

$$\mathcal{L} = \mathbb{E}_{t, \mathbf{z}_0, \boldsymbol{\epsilon}} \left[ \|\boldsymbol{\epsilon}_\theta(\mathbf{z}_t, t, \mathbf{c}) - \boldsymbol{\epsilon}\|_2^2 \right]. \tag{15}$$

**Inference process**. A latent vector $\mathbf{z}_T \sim \mathcal{N}(0, \mathbf{I})$ is randomly sampled and then refined through iterative denoising steps guided by the learned noise prediction function $\boldsymbol{\epsilon}_\theta$:

$$\mathbf{z}_{t-1} = \sqrt{\bar{\alpha}_{t-1}} \cdot \hat{\mathbf{z}}_0 + \sqrt{1 - \bar{\alpha}_{t-1}} \cdot \boldsymbol{\epsilon}_\theta(\mathbf{z}_t, t, \mathbf{c}) \tag{16}$$

where the predicted clean latent image $\hat{\mathbf{z}}_0$ is estimated by

$$\hat{\mathbf{z}}_0 = \frac{1}{\sqrt{\bar{\alpha}_t}} \left( \mathbf{z}_t - \sqrt{1 - \bar{\alpha}_t} \cdot \boldsymbol{\epsilon}_\theta(\mathbf{z}_t, t, \mathbf{c}) \right). \tag{17}$$

After completing all denoising steps, the final latent $\mathbf{z}_0$ is decoded using the VAE decoder $\mathcal{D}(\cdot)$ to reconstruct the high-resolution image $\widehat{\mathbf{X}}_0 = \mathcal{D}(\mathbf{z}_0)$. By incorporating the external conditioning signals $\mathbf{c}$ into both the reverse and inference processes, diffusion models are capable of generating images that are not only visually realistic but also semantically aligned with the target context, making them particularly effective for controlled image generation.

## A.2 PROBLEM DESCRIPTION

We denote an aerial dataset of images as $\mathcal{X} = \{\mathbf{X}_i\}_{i=1}^n$, where each $\mathbf{X}_i$ represents a high-resolution top-down view. Our goal is to develop a diffusion model over this dataset to address key challenges in generating structured and high-fidelity aerial imagery, particularly those arising from small, densely packed objects, geometric distortions, and the lack of paired text descriptions. To this end, our proposed framework enhances the generative process by integrating discrepancy-driven guidance throughout the image generation process. Specifically, we estimate discrepancy maps that capture semantic and structural inconsistencies between generations and expected outputs. These maps are leveraged in three complementary ways: (1) to assist LLMs in generating descriptive and rationale-rich text that enhances semantic conditioning, (2) to augment latent representations with spatially contextualized features that recover missing or suppressed details, and (3) to dynamically modulate the denoising process by injecting spatially localized corrective signals to refine structural details in the synthesized output.

## A.3 DETAILED STRUCTURE OF DISCREPANCY MODELING

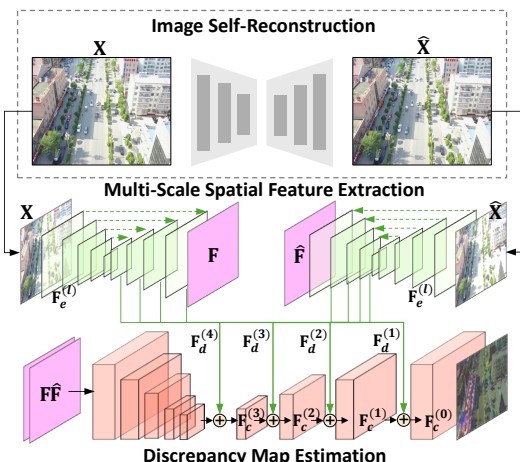

Figure 9: Discrepancy modeling that first performs image reconstruction, then extracts multi-scale spatial features, and finally estimates the discrepancy map.

The discrepancy modeling module operates in three sequential stages: (1) reconstructing the input image to expose off-manifold deviations, (2) extracting multi-scale features that preserve both global structure and local detail, and (3) fusing these features to predict a spatially coherent discrepancy map. This design enables the model to highlight error-prone regions that are critical for downstream visual reasoning and adaptive denoising.

## A.4 PROOF OF THEOREM 1

Let $\mathcal{M} \subset \mathbb{R}^{H \times W \times 3}$ be a $C^2$ embedded submanifold representing the data manifold. For any $\mathbf{X}$ sufficiently close to $\mathcal{M}$, let $\mathbf{X}_0 \in \mathcal{M}$ be the closest point under metric projection, and write the deviation as $\boldsymbol{\epsilon} = \mathbf{X} - \mathbf{X}_0$. We assume:

(A1) *Local projection*: the trained autoencoder $\mathcal{A} = \mathcal{D} \circ \mathcal{E}$ approximates the metric projection $P_{\mathcal{M}}$ to the first order near $\mathcal{M}$, such that $P_{\mathcal{M}}$: $\mathcal{A}(\mathbf{X}) = P_{\mathcal{M}}(\mathbf{X}) + o(\|\boldsymbol{\epsilon}\|)$.

(A2) *First-order linearization*: $\mathcal{A}$ is differentiable at $\mathbf{X}_0$ with Jacobian $J_{\mathcal{A}}(\mathbf{X}_0)$, so locally

$$\mathcal{A}(\mathbf{X}_0 + \boldsymbol{\epsilon}) = \mathcal{A}(\mathbf{X}_0) + J_{\mathcal{A}}(\mathbf{X}_0)\boldsymbol{\epsilon} + o(\|\boldsymbol{\epsilon}\|). \tag{18}$$

(A3) *Tangent consistency*: The Jacobian $J_{\mathcal{A}}(\mathbf{X}_0) = \Pi_{T_{\mathbf{X}_0}\mathcal{M}}$ coincides with the orthogonal projector onto the tangent space at $\mathbf{X}_0$, capturing that $\mathcal{A}$ preserves on-manifold directions to first order and contracts normal directions.

Since $\mathbf{X}_0 \in \mathcal{M}$, it follows that $\mathcal{A}(\mathbf{X}_0) = \mathbf{X}_0$. Substituting into (A2) and applying (A3), the reconstruction of $\mathbf{X}$ becomes:

$$\widehat{\mathbf{X}} = \mathcal{A}(\mathbf{X}_0 + \boldsymbol{\epsilon}) = \mathbf{X}_0 + \Pi_{T_{\mathbf{X}_0}\mathcal{M}}\boldsymbol{\epsilon} + o(\|\boldsymbol{\epsilon}\|). \tag{19}$$

Thus, the residual can be examined as:

$$\mathbf{R}(\mathbf{X}) = \mathbf{X} - \widehat{\mathbf{X}} = \boldsymbol{\epsilon} - \Pi_{T_{\mathbf{X}_0}\mathcal{M}}\boldsymbol{\epsilon} + o(\|\boldsymbol{\epsilon}\|) = \Pi_{T_{\mathbf{X}_0}\mathcal{M}^\perp}\boldsymbol{\epsilon} + o(\|\boldsymbol{\epsilon}\|). \tag{20}$$

which proves the directional statement. For the norm, decompose $\boldsymbol{\epsilon} = \boldsymbol{\epsilon}_\top + \boldsymbol{\epsilon}_\perp$ with $\boldsymbol{\epsilon}_\top \in T_{\mathbf{X}_0}\mathcal{M}$ and $\boldsymbol{\epsilon}_\perp \in T_{\mathbf{X}_0}\mathcal{M}^\perp$. Then

$$\|\mathbf{R}(\mathbf{X})\| = \|\boldsymbol{\epsilon}_\perp\| + o(\|\boldsymbol{\epsilon}\|). \tag{21}$$

By smoothness of $\mathcal{M}$ and standard properties of metric projection, we have $\text{dist}(\mathbf{X}, \mathcal{M}) = \|\boldsymbol{\epsilon}_\perp\| + o(\|\boldsymbol{\epsilon}\|)$. Combining the two expressions results in $\|\mathbf{R}(\mathbf{X})\| = \text{dist}(\mathbf{X}, \mathcal{M}) + o(\|\mathbf{X} - \mathbf{X}_0\|)$, establishing the claimed equivalence.

### A.5 PROOF OF PROPOSITION 1

Let $\{\psi_{s,k}\}$ be a Parseval frame (e.g., wavelets or Laplacian pyramid) on $\mathbb{R}^{H \times W \times 3}$ indexed by scale $s = 0, \ldots, S$ and location $k$. Any residual $\mathbf{R}$ admits a decomposition $\mathbf{R}_s = \sum_k \langle \mathbf{R}, \psi_{s,k} \rangle \psi_{s,k}$, with associated energy $E_s = \|\mathbf{R}_s\|_2^2 = \sum_k |\langle \mathbf{R}, \psi_{s,k} \rangle|^2$. By Parseval's property, the total residual energy decomposes across scales:

$$\|\mathbf{R}\|_2^2 = \sum_{s,k} |\langle \mathbf{R}, \psi_{s,k} \rangle|^2 = \sum_{s=0}^{S} E_s, \tag{22}$$

This proves the stated equality. For scale separation, low-frequency components (small $s$) are dominated by slowly varying structures such as geometric misalignments or layout shifts, while high-frequency components (large $s$) emphasize sharp features such as small-object omissions or edge artifacts. Hence, coarse scales capture global layout errors, and fine scales indicate local details. For stability under deformations, consider a smooth geometric deformation $\tau$ with $\|\nabla\tau - I\|_\infty \le \kappa \ll 1$. For wavelet-type frames with $C^1$ generators, deformation stability results imply

$$\sum_k |\langle \mathbf{R} \circ \tau, \psi_{s,k} \rangle - \langle \mathbf{R}, \psi_{s,k} \rangle|^2 \le C\kappa^2 2^{2s} \|\mathbf{R}\|_2^2, \tag{23}$$

so the subband energies $E_s$ vary smoothly under small perturbations of the input. This ensures robust detection of global discrepancies at coarse scales and local omissions at fine scales.

### A.6 PROOF OF PROPOSITION 2

At early timesteps, the latent $\mathbf{z}_t$ is heavily corrupted (large $\sigma_t$), so intermediate reconstructions are unreliable and the estimated discrepancy $\mathbf{D}_t$ is noisy and biased. Using it to steer denoising risks is more likely to inject artifacts than to improve alignment. As noise decays (small $\sigma_t$), reconstructions stabilize; $\mathbf{D}_t$ concentrates around the true residual and becomes informative for targeting normal-space errors such as missing vehicles or misaligned roads. Hence, discrepancy guidance is most effective late in the reverse process.

Formally, let $L_t(\mathbf{z}) = \mathbb{E}\|\boldsymbol{\epsilon} - \boldsymbol{\epsilon}_\theta(\mathbf{z}, t, \mathbf{c}_t)\|_2^2$ denote the per-timestep denoising risk. Consider a single-step update $\mathbf{z}_t^+ = \mathbf{z}_t + \eta\, u(\mathbf{D}_t)$ with step size $\eta > 0$ and a direction $u(\cdot)$ aligned with the normal component of the denoising model's error gradient. A first-order expansion yields

$$L_t(\mathbf{z}_t^+) \approx L_t(\mathbf{z}_t) + \eta \langle \nabla_{\mathbf{z}} L_t(\mathbf{z}_t), u(\mathbf{D}_t) \rangle + O(\eta^2). \tag{24}$$

Decompose the estimated discrepancy as $\mathbf{D}_t = \mathbf{D} + b_t + n_t$, where $\mathbf{D}$ is the true (noise-free) discrepancy signal, the systematic bias satisfies $\|b_t\| = \mathcal{O}(\sigma_t)$, and the zero-mean noise $n_t$ satisfies $\mathbb{E}\|n_t\|^2 = \mathcal{O}(\sigma_t^2)$. Taking expectation and using alignment $\langle \Pi_{\text{normal}} \nabla_{\mathbf{z}} L_t, u(\mathbf{D}) \rangle \le -\kappa \|\Pi_{\text{normal}} \nabla_{\mathbf{z}} L_t\|_2^2$ for some $\kappa > 0$, we obtain

$$\mathbb{E}\big[L_t(\mathbf{z}_t^+) - L_t(\mathbf{z}_t)\big] \le -\eta\,\kappa \|\Pi_{\text{normal}} \nabla_{\mathbf{z}} L_t\|_2^2 + \eta\,\mathcal{O}(\sigma_t) + \mathcal{O}(\eta^2). \tag{25}$$

Thus, for sufficiently small $\sigma_t$ (late timesteps) and small enough $\eta$, the negative descent term dominates, guaranteeing a decrease in expected denoising risk. When $\sigma_t$ is large (early timesteps), the $\mathcal{O}(\sigma_t)$ term can dominate, and no improvement is guaranteed. This establishes the existence of a threshold $\sigma^\star$ below which discrepancy-guided updates are beneficial.

## A.7 TRADITIONAL VISUAL REASONING PROMPT

Traditional visual reasoning prompts like $P^{\text{visual}}$ Mao et al. (2023), rely solely on the input image $\mathbf{X}_i$ and focus on salient regions without capturing localized inconsistencies, failing to offer targeted feedback essential for refining complex aerial scenes.

> $P^{\text{visual}}$: Generate a concise rationale that interprets an *[aerial image]*. Start with high-level context (lighting and time of day) to frame visibility and activity. Then reason through the scene region by region, analyzing how objects behave and interact. Highlight uncovering patterns like flow, clustering, or contrast to reveal deeper insights into the scene's structure.

## A.8 USE OF LARGE LANGUAGE MODELS (LLMS)

LLMs were used solely to assist in language refinement and copyediting of portions of the manuscript. All core research contributions, including conceptual development, methodology design and implementation, and experimental results, were conducted entirely by the authors. No LLMs were used to generate code, raw data, or figures. However, because optimizing actor-critic visual reasoning is a core contribution of this work, LLMs were intentionally used as part of the proposed methodology to generate text descriptions and rationales for aerial image datasets and to validate the effectiveness of the designed reasoning strategy. This constitutes a technical contribution and is explicitly detailed in the main content of the paper.

