# OpenReview forum: "Mind the Gap: Discrepancy-Centric Diffusion for Aerial Image Reasoning and Synthesis"
_ICLR.cc/2026/Conference — ICLR 2026 Conference Withdrawn Submission_

### Official Review · Reviewer_jteJ · 2025-10-29

**Soundness:** 2
**Presentation:** 3
**Contribution:** 2
**Rating:** 4
**Confidence:** 3

**Summary:**

The paper introduces the MIND framework for text-guided aerial image synthesis. The method aims to improve spatial alignment, geometric fidelity, and semantic consistency in complex aerial imagery by incorporating discrepancy maps. Experiments on VisDrone-DET and DroneRGB-T show improved FID, PSNR, and LPIPS scores compared to several diffusion-based baselines.

Overall, I find the paper practically valuable, addressing a domain with clear real-world relevance. However, there are still some concerns regarding efficiency and generalization.

**Strengths:**

* The paper accurately identifies the specific challenges of aerial image generation and tailors its solution accordingly.
* The integration of multi-scale discrepancy modeling with actor–critic prompting and adaptive denoising is well thought out and technically ambitious.
* The experimental section is broad, covering multiple datasets and ablation studies that consistently show quantitative and qualitative improvements over baselines.

**Weaknesses:**

* **Inference efficiency.** The paper reports using DDIM with 300 denoising steps and a guidance scale of 8.0, which is significantly higher than typical inference settings. This raises concerns about practical deployment speed and scalability.
* **Resolution generalization.** It is not evident whether the approach generalizes beyond the tested resolution of 512×512, especially for higher-resolution aerial imagery where geometric fidelity is critical.
* **Reliance on external reasoning.** The “actor–critic” component depends on GPT-4 to generate semantic cues, which makes it difficult to isolate how much of the improvement comes from the proposed MIND framework versus the LLM’s inherent semantic prior.

**Questions:**

* Could the authors clarify the inference cost relative to standard diffusion models and baselines, including working time for each module?
* Is it feasible to apply the proposed approach to few-step diffusion models (e.g., FLUX-Schnell*) or advanced solvers (e.g., DPM++*) to better represent realistic practical scenarios?
* How does the model’s performance vary with different numbers of inference steps? Does the discrepancy guidance remain stable under aggressive step reduction?

*https://huggingface.co/black-forest-labs/FLUX.1-schnell

*DPM-Solver++: Fast Solver for Guided Sampling of Diffusion Probabilistic Models

---

### Official Review · Reviewer_3NX2 · 2025-10-30

**Soundness:** 2
**Presentation:** 2
**Contribution:** 2
**Rating:** 4
**Confidence:** 5

**Summary:**

The paper introduces MIND (MultI-scale discrepaNcy-centric latent Diffusion), a framework for high-fidelity, semantically coherent text-to-aerial image synthesis. It's designed to overcome the unique challenges of aerial imagery, such as high object density, geometric distortions, and the scarcity of paired text-aerial image data. The core innovation is the use of discrepancy maps to identify semantic and structural inconsistencies during image generation. It has a multi-scale module which estimates discrepancy maps from autoencoder reconstruction residuals, effectively highlighting off-manifold structural deviations and hard-to-learn regions. An actor-critic prompting paradigm uses Large Language Models (LLMs) where the Actor generates an initial, global semantic description. The Critic uses the discrepancy map to refine the description with rationale-rich annotations, correcting omissions and enhancing semantic grounding in uncertain regions. The discrepancy map is projected and injected into the latent representation via spatial attention-modulated fusion to restore fine-grained details lost during compression. The denoising process is dynamically steered by continuous discrepancy feedback, updating the latent features and multi-modal conditioning vector to refine structural details in error-prone regions, primarily in later timesteps where noise is low.

**Strengths:**

1. The paper demonstrates SOTA performance on various metrics.
2. The core idea of using discrepancy maps provides a principled way to target and correct errors, which is crucial for complex aerial scenes.
3. The actor-critic visual reasoning addresses the lack of high-quality paired text-aerial image data by leveraging LLMs to generate both global scene descriptions and localized, error-aware rationales.

**Weaknesses:**

The entire discrepancy-centric approach relies on the autoencoder producing reliable reconstructions and residuals. A poor autoencoder could lead to inaccurate discrepancy maps, degrading the effectiveness of the guiding signals.

The framework requires training and integrating multiple complementary components (multi-scale feature extraction networks, discrepancy map estimation network, and an adaptive denoising mechanism) in addition to the base latent diffusion model, increasing overall complexity and computational overhead.

**Questions:**

Given the multi-scale feature extraction, discrepancy map calculation, and adaptive denoising steps at later timesteps, how does the inference time of MIND compare to state-of-the-art baselines like DiffusionSAT or Stable Diffusion?

Does the discrepancy modeling module generalize well to errors generated during the denoising process of a completely random latent vector, which may present different types of inconsistencies than those from an autoencoder reconstruction?

The paper mentions using GPT-4o for actor-critic reasoning. How robust is the final model's performance to using a smaller, less powerful LLM for the reasoning component, particularly concerning the rationale generation?

Proposition 2 motivates applying adaptive denoising only after a certain t. Is there a detailed analysis or visualization that shows the effect of the threshold or the transition point on the quality of generated images

The code is missing, can you please provide the code?

More qualitative results and qualitative results for ablations will be useful to evaluate the effectiveness of the proposed method.

---

### Official Review · Reviewer_gKuf · 2025-10-31

**Soundness:** 2
**Presentation:** 2
**Contribution:** 1
**Rating:** 2
**Confidence:** 4

**Summary:**

This paper proposes MIND, a latent diffusion framework designed to improve the quality of aerial image synthesis. The authors identify challenges specific to aerial imagery, such as high object density and geometric distortions. The core of their method is the use of a "discrepancy map," which is generated by calculating the reconstruction error from an autoencoder. This map is intended to identify semantic and structural inconsistencies.

This discrepancy map is then leveraged in three main ways:

1. Actor-Critic Visual Reasoning: The map is fed to an LLM (GPT-40) in a two-step process. An "Actor" generates a scene description, and a "Critic" uses the discrepancy map to identify "inconsistencies" and refine the text.

2. Latent Augmentation: The discrepancy signal is injected into the latent representation.

3. Adaptive Denoising: During the reverse diffusion process, the discrepancy map is re-calculated at later timesteps to dynamically correct the latent vector and conditioning signals.

The authors evaluate their method on the VisDrone-DET and DroneRGB-T datasets, reporting superior performance over several baselines using FID, PSNR, KID, and LPIPS metrics.

**Strengths:**

The paper tackles a relevant and challenging problem. Generating high-fidelity, large-scale aerial images is difficult, and the specific issues (small objects, high density, distortion) are well-identified.

**Weaknesses:**

My primary concerns with this work are related to a fundamentally flawed evaluation setup, which makes it impossible to fairly assess the paper's contributions, and a significant overstatement of the core technical novelty.

1. Fundamentally Unfair Evaluation: The entire quantitative evaluation appears to be invalid due to an apples-to-oranges comparison. The proposed method, MIND, is an image-conditioned model. It uses the source image as a "spatial prior" (Section 3.3, "Conditioning vector construction") which is encoded via BLIP. However, the baselines it is compared against (e.g., Stable Diffusion, ARLDM) are primarily text-to-image (T2I) models. It is not stated that these baselines were given the same image-conditioning (e.g., as an img2img or ControlNet-style input).

- If the baselines only received text, it is completely unsurprising that the proposed image-conditioned method achieves better PSNR, LPIPS, and FID to the source image. These metrics would simply be rewarding the model for effectively copying its input, not for its generative quality. This is not a fair comparison and does not support the paper's claims.

2. Inappropriate Choice of Metrics: Following the point above, the paper is framed as a generative synthesis model, but it is exclusively evaluated using reconstruction or translation metrics. PSNR, LPIPS, and KID (when used this way) measure fidelity to a single ground-truth image. They do not measure the two key aspects of a generative model: quality and diversity.

- There is no evaluation of generative diversity (e.g., are the outputs just minor variations of the input?).

- A much more convincing evaluation would have been to measure the utility of the generated images in a downstream application (e.g., training an object detector) or, at minimum, conduct a human study to assess realism and fidelity. As it stands, the quantitative results are not meaningful for a generative task.

3. Overstated Novelty of "Discrepancy Map": The paper provides a theoretical justification (Theorem 1) for the discrepancy map, framing it as a sophisticated tool for identifying "off-manifold" and "normal-space" deviations. However, the actual implementation is just the $L_1$ pixel-wise difference between an image and its autoencoder reconstruction ($D_{gt} = |X-\hat{X}|$).

- This is a simple pixel-error map, not necessarily a "semantic and structural" inconsistency map. It is well-known that autoencoders struggle to reconstruct high-frequency details. This "discrepancy map" is therefore likely just highlighting trees, water, and building edges, not high-level semantic errors like a missing car or a misplaced building. This weak foundation undermines the subsequent reasoning module.

4. Reproducibility and Robustness of the LLM-Reasoner: A core contribution, the "Actor-Critic" reasoning, is entirely dependent on a single, massive, proprietary model (GPT-40). This presents a serious barrier to reproducibility and raises questions about robustness.

- The authors provide no analysis of whether this component is essential or if it works with smaller, open-source models (e.g., LLaMA, Mistral). If the method's performance collapses without this specific black-box API, its scientific contribution is questionable.

- Furthermore, it's completely vague how the pixel-error map (a 3xHxW tensor) is "interpreted" by the critic LLM to find "inconsistencies." This entire process feels like hand-waving.

5. Unjustified Complexity: The full MIND system is incredibly complex, requiring a VAE for latent diffusion, a second autoencoder for discrepancy, a multi-scale U-Net for discrepancy modeling, an LLM API for reasoning, and a modified denoising loop. The ablation study itself (Table 3) shows that the adaptive denoising (D-Denoising) adds only a small improvement over the other components, suggesting this added complexity may not be justified.

**Questions:**

1. Can the authors please clarify the exact inputs provided to all baseline models in Table 1? Did "Stable Diffusion," "ARLDM," etc., receive only text, or were they also conditioned on the source image in a fair img2img or ControlNet-style setup?

2. If the baselines were text-only, how can the authors justify claiming this is a fair comparison for a generative task using reconstruction metrics?

3. Why were no standard generative metrics (e.g., Inception Score, or distribution-based Precision/Recall) or downstream task evaluations used to measure the quality and diversity of the generated images, rather than just their similarity to the input?

4. Can the authors provide a concrete example of the prompt used for the "Critic" LLM? How is the raw, pixel-level discrepancy map ($D$) converted into a form the LLM can "reason" about to find "spatial coherence" issues?

5. Have the authors tested the Actor-Critic module with any model other than GPT-40? How critical is this specific proprietary model to the method's performance?

**Details Of Ethics Concerns:**

I am flagging this paper for a research integrity review due to what appears to be a clear case of dual submission.

This ICLR 2026 submission, titled "MIND THE GAP: DISCREPANCY-CENTRIC DIFFUSION FOR AERIAL IMAGE REASONING AND SYNTHESIS," is substantially identical to a paper submitted to WACV 2026 (Submission 760), titled "Diffusion with Correction Signals: Discrepancy-Aware Reasoning via Text-Vision Interaction for Aerial Image Synthesis."

Both papers propose the same core methodology:

1. Using "multi-scale discrepancy maps" to identify "semantic and structural inconsistencies."

2. Employing an "Actor-Critic" (ICLR) or "Actor-Critic-Solver" (WACV) reasoning paradigm based on an LLM to interpret these maps.

3. Using "adaptive denoising" to apply corrections during the generation process.

4. The pipeline figures are exactly the same.

Furthermore, both papers are evaluated on the exact same datasets (VisDrone-DET and DroneRGBT) and address the identical problem. The contributions, methodology, and experiments are functionally the same, with only superficial changes to project names ("MIND" vs. "Rectify") and phrasing. This appears to be a clear violation of the ICLR dual submission policy.

---

### Official Review · Reviewer_yyK5 · 2025-10-31

**Soundness:** 2
**Presentation:** 3
**Contribution:** 2
**Rating:** 4
**Confidence:** 2

**Summary:**

The paper proposes MIND—a MultI-scale discrepaNcy-centric latent Diffusion framework for aerial image synthesis. The key idea is to estimate discrepancy maps from autoencoder reconstruction residuals and use them to (i) localize spatial/semantic inconsistencies, (ii) drive an actor–critic visual reasoning pipeline to produce rationale-rich textual guidance, (iii) augment latents with spatial corrections, and (iv) perform adaptive denoising that focuses later diffusion steps on “hard” regions. Experiments on VisDrone-DET and DroneRGB-T report consistent improvements over general-purpose and aerial-specific diffusion baselines across FID, PSNR, KID, LPIPS; ablations attribute gains to each module and show the actor–critic prompting improves CLIP-score/alignment.

**Strengths:**

+ The approach directly targets aerial‑specific challenges (object density, occlusion, scale distortions) and proposes a concrete conditioning/denoising schedule to tackle the problem.

+ The estimator uses standard encoders/decoders, attention gating (Eq. 5–6), and a straightforward loss (Eq. 7–8), making reproduction feasible for practitioners.

**Weaknesses:**

+ Section 3.3 starts from an input image $X$ encoded to $z_0$ then denoised, suggesting an image‑conditioned (I2I) setup rather than pure T2I. It is unclear whether baselines were run in comparable image‑conditioned modes; if text‑only baselines were used, comparisons may overstate gains. (p. 5; Fig. 1 also shows an explicit “Input Image” path).

+ The abstract claims significant improvements; the main text (as provided) lacks human evaluations, and robustness checks (lighting extremes, viewpoint shifts).

+ Sensitivity to hyper-parameters, attention‑fusion vs. simple concat, and when/how many timesteps get discrepancy guidance is not fully explored

**Questions:**

+ Were competing methods run in image‑conditioned settings (e.g., I2I or ControlNet‑style layout) to match MIND? If not, please add such baselines and report both text‑only and image‑conditioned comparisons.

+ What is the impact of replacing attention‑based fusion (Eq. 5–6) with additive/concat fusion? How sensitive are outcomes to the hyper-parameters ?

---

### Note · Authors · 2025-11-12

I have read and agree with the venue's withdrawal policy on behalf of myself and my co-authors.